# Determinants of solid fuel use in Sub-Saharan Africa: A multilevel analysis using DHS data

**Jember Azanaw** *, **Mastewal Endalew**

Department of Environmental and Occupational Health and Safety, Institute of Public Health, College of Medicine and Health Sciences, University of Gondar, Gondar, Ethiopia

* jemberazanaw21@gmail.com

## Abstract

### Background

Dawn in human history life excessively depends on different energy sources for various purposes including cooking food and heating. Energy sources determine the economic development of the community, at the same time it is a public health problem due to environmental pollution. Worldwide Health Organization (WHO) data indicate that about 7 million deaths are attributed to indoor air pollution yearly. According to the 2019 WHO report, 90% of African people depend on dirty energy sources for domestic purposes. Understanding the prevalence and factors of solid fuel use enables policymakers to take measures to prevent its effect on public health by concerned bodies. However, research is done covering such a large area, and the sample size is limited. Therefore, the main objective of this research is to determine factors affecting solid fuel use in Sub-Saharan Africa.

### Methods

The data source is the Demographic and Health Surveys (DHS), a regionally representative survey. Households in DHS are selected using a two-stage cluster sampling methodology. A total of 233, 391 weighted samples were included in the study. A multilevel logistic regression modeling approach was applied to estimate the influence of both individual and community-level factors on solid fuel use.

### Results

The prevalence of solid fuel use in sub-Saharan Africa was 82.05%, with 95% CI (81.90, 82.21). Based on multilevel regression of the final model, household heads aged over 60 years (AOR = 1.12, 95% CI; 1.05–1.19), unmarried household heads (AOR = 1.14, 95% CI; 1.09–1.20), household heads no having education (AOR = 4.91, 95% CI; 4.59–5.25), poor wealth index (AOR = 12.46, 95% CI; 11.34–13.70), not watching television (AOR = 2.38, 95% CI; 2.23–2.53), households without access

**Data availability statement:** The data utilized in this study were obtained from the Demographic and Health Surveys (DHS) Program. These datasets are publicly available but require registration and approval from the DHS Program Data Archivist. Researchers can request access by creating an account and submitting a data request through the DHS Program website at https://dhsprogram.com/data/dataset_admin/login_main.cfm. Upon approval, the relevant datasets can be accessed for research purposes.

**Funding:** The author(s) received no specific funding for this work.

**Competing interests:** The authors have declared that no competing interests exist.

**Abbreviations:** UN: United Nations, WHO: World Health Organization, UNICEF: United Nations Children's Fund, SSA: sub-Saharan Africa, AIC: Akaike's Information Criterion, AOR: Adjusted Odds Ratio, CI: Confidence Interval, DHS: Demographic and Health Survey, DHS: Demographic and Health Survey, ICC: Intra Class Correlation, MOR: Median Odd Ratio, PCV: Proportional Change in Variance

to electricity (AOR = 3.97, 95% CI; 3.74–4.22), Family size between four & seven (AOR = 3.94, 95% CI; 3.68–4.21), lower education levels (AOR = 2.18, 95% CI; 1.84–2.59), low media exposure (AOR = 1.39, 95% CI; 1.30–1.49), low-income levels (AOR = 63.42, 95% CI; 56.15–71.62) and being rural (AOR = 3.88, 95% CI; 3.68–4.09) were significantly associated with solid fuel use in sub-Saharan Africa.

## Conclusions

The study showed that the prevalence of solid fuel use was high in sub-Saharan Africa. Factors such as the age and marital status of the household head, educational status of the household head, wealth index, watching television, access to electricity, family size, community-level education, community-level media exposure, residence, income level, and community-level poverty were significantly associated with solid fuel use. Solid fuel cooking has been related to respiratory and cardiovascular problems such as lung cancer, chronic obstructive pulmonary disease, and heart disease, as well as pneumonia in children, the elderly, and women who spend most of their time at home. Public health policymakers and possible stakeholders should act on poverty reduction, increase access to electricity, and educational messages through media, and increase educational infrastructures, which can minimize solid fuel use. It is also vital to raise awareness of the potential health risks related to solid fuel use in sub-Saharan Africa. Since human beings exist on land, life excessively depends on different forms of energy sources for various purposes including cooking food and heating sources. Energy sources determine the economic development of the community, at the same time it is a public health problem due to environmental pollution [1,2]. Especially using dirty energy sources like cow dung, firewood, crop residue, and charcoal are sources of indoor air pollution leading to health problems for inhabitants [3]. Worldwide Health Organization (WHO) data indicate that about 7 million deaths are attributed to indoor air pollution each year; millions more are at risk of heart attacks, lung ailments, strokes, and other respiratory and cardiovascular conditions [4]. Reports indicate that household air pollution causes an expected 1.6–3.8 million premature deaths annually [5,6].

## Background

Firewood, charcoal, straw/shrubs/grass, animal dung, coal, lignite, kerosene, and crop residue are under the category of traditional energy sources [7–9]. The reason why kerosene is categorized as unclean fuel is that it produces harmful levels of household air pollutants and safety risks for users from handling and accidental fires [10]. Even if using clean fuel for domestic purposes has multidimensional public health benefits, it is difficult to access it due to poverty, and lack of awareness, in the developing world [11]. In a 2020 joint report by the UNSD, World Bank, and WHO, 3 billion people who used solid fuel were found in low and middle-income countries [12]. According to 2019, WHO reports that 90% of African people depend on dirty

energy sources for domestic purposes [6,13]. Despite there being development among countries including clean fuel use, countries in sub-Saharan Africa still rely on solid fuel use as a source of energy [14]. The use of firewood as fuel is persistent in the region, which is 2–3 times more likely compared with other parts of the world [15].

Earlier similar studies conducted in different parts of the world indicated that the educational level of household heads, wealth status of households, residence type [16–18], number of family sex of household heads, age of household head, and marital status were determinants of solid fuel use [19,20]. Many studies were conducted in the region, which focused on the prevalence, predictors, and trends, including spatial analysis. However, these studies were based on using a small sample size or specific location [17,21] or a single demographic health survey (DHS) [19,22,23] or including a few countries [14] unable to represent SSA. Hence, research is done covering such a large area, and the sample size is limited. Hence, such a larger dataset enhances the generalizability and robustness of the findings. Therefore, the main objective of this study is to determine factors affecting Solid Fuel Use in sub-Saharan Africa based on 29 countries' DHS data.

The determinants of solid fuel use in Sub-Saharan Africa are of significant interest due to their impact on health, environmental sustainability, and socio-economic conditions in the region. Understanding these factors through a multilevel analysis using DHS data can provide valuable insights for targeted interventions and policy development to reduce health risks associated with solid fuel use.

## Method

### Study area and data source

Sub-Saharan African (SSA) countries represent a diverse range of socioeconomic, cultural, and geographic contexts for analyzing solid fuel use comprehensively and regionally. The data source is the Demographic and Health Surveys (DHS), a nationally representative survey that collects data on various health and population metrics, including household energy sources. The DHS data used in this study include multiple years, enabling an examination of changes over time and the identification of persistent and emerging trends in solid fuel use across SSA. Data collection followed standardized procedures, including stratified sampling to ensure regional representation within each country. Households in DHS are selected using a two-stage cluster sampling methodology. In the first stage, cluster enumeration areas (EAs) were sampled using a probability proportional to the population size technique. In the second stage, all households in the selected area were chosen at random for interviews. This sampling strategy was utilized to get a representative sample of households. The authors corrected for survey year and country effects using fixed effects or stratification as necessary, normalized variables to guarantee consistency across surveys, and considered survey-specific sampling weights to reflect population-level estimates. By leveling the contributions from each survey and performing sensitivity analyses to verify the robustness of our findings, we have also addressed potential biases resulting from variations in dataset sizes. These methodological principles guarantee that the validity of the study is not jeopardized by merging datasets. A total of 233391 weighted samples were included in the study.

### Study variables

**Outcome variable.** The outcome variable was the fuel type used for a different purpose at the household level. Cooking fuels such as wood, charcoal, straw/shrubs/grass, animal dung, coal, lignite, kerosene, and crop residue/wood were categorized as 'solid fuels', while natural gas, liquefied petroleum gas (LPG), biogas, and electricity were classified as 'clean fuels [24]. Then, fuel types were categorized as '1' for 'solid fuel' and '0' for 'clean fuel.'

### Predictor variables

**Individual and community-level factors.** The study analyzes both individual-level factors (household size, education, wealth index, sex, and age of household head) and community-level factors (urban/rural status, community-level media exposure, community-level education, and region). These variables are extracted and harmonized across all 33 countries to facilitate a consistent analysis. The proportion of households is categorized as low poverty level (those with ≥ 50%)

and higher poverty level (those with <50%) using the mean value. The proportion of households is categorized as high community-level education (those with ≥ 50%) and low community-level education (those with <50%) using the mean value to minimize bias and maintain the robustness of the analysis. Missing data were treated based on imputation, and four DHSs with no appropriate data were recalled from the analysis.

## Data analysis

Because it takes into consideration the data's hierarchical structure in which specific individual-level factors are nested within more general contextual units like communities, we used the multilevel logistic regression model [25,26]. This method addresses any clustering effects that can cause skewed standard errors in single-level models while capturing differences in solid fuel use selection at the individual and community levels. In keeping with the goal of the study, multilevel modeling also allows us to investigate how contextual factors affect household decisions outside of individual socioeconomic and demographic features. Therefore, this study accounts for the hierarchical structure of the data, where households are nested within communities and countries, making a multilevel analysis essential to understanding variations in solid fuel use at different levels. Hence, a multilevel logistic regression modeling approach was applied to estimate the influence of both individual and community-level factors on solid fuel use. The multilevel logistic regression model is structured in four levels.

The display of P-values and Adjusted Odds Ratios (AOR) can indeed be ambiguous, as their interpretation depends on the context of the study and the statistical model used. AOR represents the strength and direction of the association between an exposure and an outcome, adjusted for other variables in the model. High or low odds ratios often indicate a strong association between the factor and the outcome. An extremely high AOR suggests that the factor significantly increases the likelihood of the outcome, while a very low AOR implies a protective effect or a strong negative association.

Model selection in statistical analysis was guided by criteria such as the Akaike Information Criterion (AIC), the Bayesian Information Criterion (BIC), and the Intraclass Correlation Coefficient (ICC). AIC and BIC balance model fit and complexity, with lower values indicating a better trade-off. AIC tends to favor more complex models, while BIC penalizes complexity more heavily, often leading to simpler models. ICC, on the other hand, is used in multilevel modeling to assess the proportion of variance attributable to higher-level groupings. A high ICC suggests that a significant portion of the variability in the data is due to these groupings, justifying the use of multilevel models. Together, these criteria helped us select the most appropriate model, ensuring that the results are both statistically sound and interpretable.

An empty model (Model 0) was the intercept-only model, without a dependent variable.

$$\log\left(\frac{P}{1-P}\right) = \beta 0 + uj$$

Where:

P: the probability of the outcome.

β0: intercept.

uj: random effect for the j-th cluster

Model I: was adjusted for individual-level predictors in fuel use prediction.

$$\log\left(\frac{P}{1-P}\right) = \beta 0 + \beta 1(\textit{Sex of HHH}) + \beta 2(\textit{Age of HHH}) + \beta 3(\textit{Marital status})$$
$$+ \beta 4(\textit{Educational status}) + \beta 5(\textit{Wealth index}) + \beta 6(\textit{Watching television})$$
$$+ \beta 7(\textit{Electricity access}) + \beta 8(\textit{Number of family members}) + uj$$

Where:

β1, β2, β3, β4, β5, β6, β, β8: coefficients for each individual-level variable.

uj: random effect for clustering.

Model II: was accustomed to incorporating community-level variables.

$$\log\left(\frac{P}{1-P}\right) = \beta0 + \beta9(\textit{Community education level}) + \beta10(\textit{Community media exposure})$$
$$+ \beta11(\textit{Community poverty level}) + \beta12(\textit{Type of residence}) + \beta13(\textit{Region}) + uj$$

Where:

β9, β10, β11, β12, β13: coefficients for each community-level variable.

uj: random effect for clustering.

Model III: was adjusted for integrating both individual and community-level factors.

$$\log\left(\frac{P}{1-P}\right) = \beta0 + \beta1(\textit{Sex of HHH}) + \beta2(\textit{Age of HHH}) + \beta3(\textit{Marital status})$$
$$+ \beta4(\textit{Educational status}) + \beta5(\textit{Wealth index}) + \beta6(\textit{Watching television})$$
$$+ \beta7(\textit{Electricity access}) + \beta8(\textit{Number of family members})$$
$$+ \beta9(\textit{Community education level}) + \beta10(\textit{Community media exposure})$$
$$+ \beta11(\textit{Community poverty level}) + \beta12(\textit{Type of residence})$$
$$+ \beta13(\textit{Region}) + uj$$

### Random Effect and Model Fit Assessment

Model fit is evaluated using several criteria, including the Intraclass Correlation Coefficient (ICC) to measure the proportion of variance explained at the community level, Median Odds Ratio (MOR) for assessing heterogeneity between communities, Proportional Change in Variance (PVC) to indicate changes in variance between models, and Bayesian Information Criterion (BIC) and Akaike Information Criterion (AIC) for model selection. Additionally, the Deviance Information Criterion (DIC) is employed to compare the fit between models. The empty model serves as a baseline, followed by Models I, II, and III, which sequentially add more complexity and contextual factors. Lower values of BIC, AIC, and DIC indicate the best model fit [27], while the ICC and MOR provide insight into the relative importance of community-level factors in influencing solid fuel use.

### Ethics approval and consent to participate

It is not applicable (because it is secondary data). However, during primary data collection by DHS participants, anonymity was protected, and the ethical requirements were adhered to.

### Results

#### Socio-Demographic Characteristics

There were 233 391 weighted households involved in this study. In this study, 48.65% of household heads (113,552) were aged between 35 and 60. Among household heads, about 168,831 (72.34%) were males, and 92,039 (39.44%)

were living in rural areas. The majority **(**57.78%) of the households found in sub-Saharan African countries were without the accessibility to electricity. Nearly sixty percent (60.56%) of the included households were found to be rural residents, and 35.52% of them did not watch television. Only 11.03% (23,714/233,372) of the incorporated households were in upper-middle-income status, whereas the remaining were low-income (58.27%) and lower middle income (30.70%). East Africa accounts for 102,856 (44.07%), followed by West Africa 90,278 (38.68%).(Table 1).

## Solid fuel use prevalence

The overall prevalence of Solid fuel use among Sub-Saharan African countries was 82.05% (81.90, 82.21%). Countries such as Nigeria (10%), Malawi (9.3%), and Burundi (6.73%) have relatively high proportions of solid fuels used for

**Table 1. Overview of Socio-demographic Characteristics at the Individual and Community Levels among Households in Sub-Saharan African Countries Based on Data from 233,391 Observations.**

| Variable | Categories | Freq. | Percent |
|---|---|---|---|
| Sex of HHH | Male | 168,831 | 72.34 |
| | Female | 64,560 | 27.66 |
| Number of Household Members | <4 | 83,079 | 35.60 |
| | 4-7 | 113,838 | 48.78 |
| | > 7 | 36,474 | 15.63 |
| Accessibility of Electricity | Yes | 98,532 | 42.22 |
| | No | 134,859 | 57.78 |
| Type of Residence | Urban | 92,039 | 39.44 |
| | Rural | 141,352 | 60.56 |
| Marital Status | Married | 165,550 | 72.65 |
| | Unmarried | 62,339 | 27.35 |
| Age of Head of Household | <35 | 77,179 | 33.07 |
| | 35-60 | 113,552 | 48.65 |
| | >60 | 42,660 | 18.28 |
| Watching Television | No | 150,491 | 64.48 |
| | Yes | 82,885 | 35.52 |
| Education Status of HHH | No education | 75,102 | 32.19 |
| | Primary | 73,847 | 31.65 |
| | Secondary | 61,791 | 26.49 |
| | Higher | 22,549 | 9.67 |
| Wealth Index | Poor | 84,116 | 36.04 |
| | Middle | 45,454 | 19.48 |
| | Rich | 103,821 | 44.48 |
| Community Level Education | Low | 116,215 | 49.79 |
| | High | 117,176 | 50.21 |
| Country Income Status | Low income | 125,334 | 58.27 |
| | Lower middle income | 66,037 | 30.70 |
| | Upper middle income | 23,714 | 11.03 |
| Community-Level Media Exposure | Unexposed | 86,938 | 37.25 |
| | Exposed | 146,434 | 62.75 |
| Region | East Africa | 102,856 | 44.07 |
| | Central Africa | 21,705 | 9.30 |
| | West Africa | 90,278 | 38.68 |
| | Southern Africa \| | 18,552 | 7.95 |

domestic purposes. Conversely, Lesotho (0.06%), South Africa (0.61%), and Liberia (0.66%) exhibit much lower proportions of solid fuel use (Fig 1).

## Multilevel logistic regression modeling for both individual and community-level factors

Variables with a P-value <.25 in the bivariable analysis were selected for multivariate analysis to reduce the risk of omitting potentially significant predictors. Among individual-level factors, the age, the marital status of the household head, the educational status of the household head, the wealth index, watching television, access to electricity, and family size were significantly associated with solid fuel in the first model. In the second model, community-level education, community-level media exposure, residence, income level, and community-level poverty were significantly associated with solid fuel use. All of the characteristics that were statistically significant in the first and second models were associated with household solid fuel usage in the final model.

Based on regression results, households headed by individuals aged over 60 had 1.12 times higher odds of solid fuel use (AOR = 1.12, 95% CI: 1.05, 1.19) compared to those headed by individuals under 35. This possibly reflects the generational persistence of traditional fuel use patterns and limited adoption of modern energy sources among older individuals.

Unmarried household heads are 1.14 times more likely to use solid fuels compared to married ones (AOR = 1.14, 95% CI: 1.09, 1.20). This may be due to differences in resource allocation or decision-making dynamics in unmarried versus married households.

Similarly, households with heads having no education, primary education, or secondary educational statuses were 4.91 (AOR = 4.91, 95% CI: 4.59, 5.25), 3.96 (AOR =3.96, 95%CI: 3.74, 4.20), 1.14 (AOR = 1.14, 95%CI: 1.09, 1.20) times higher odds of using solid fuel compared to those with higher education, respectively. This revealed the critical role of education in driving energy transitions, as better-educated individuals might have greater awareness and resources to adopt cleaner energy alternatives.

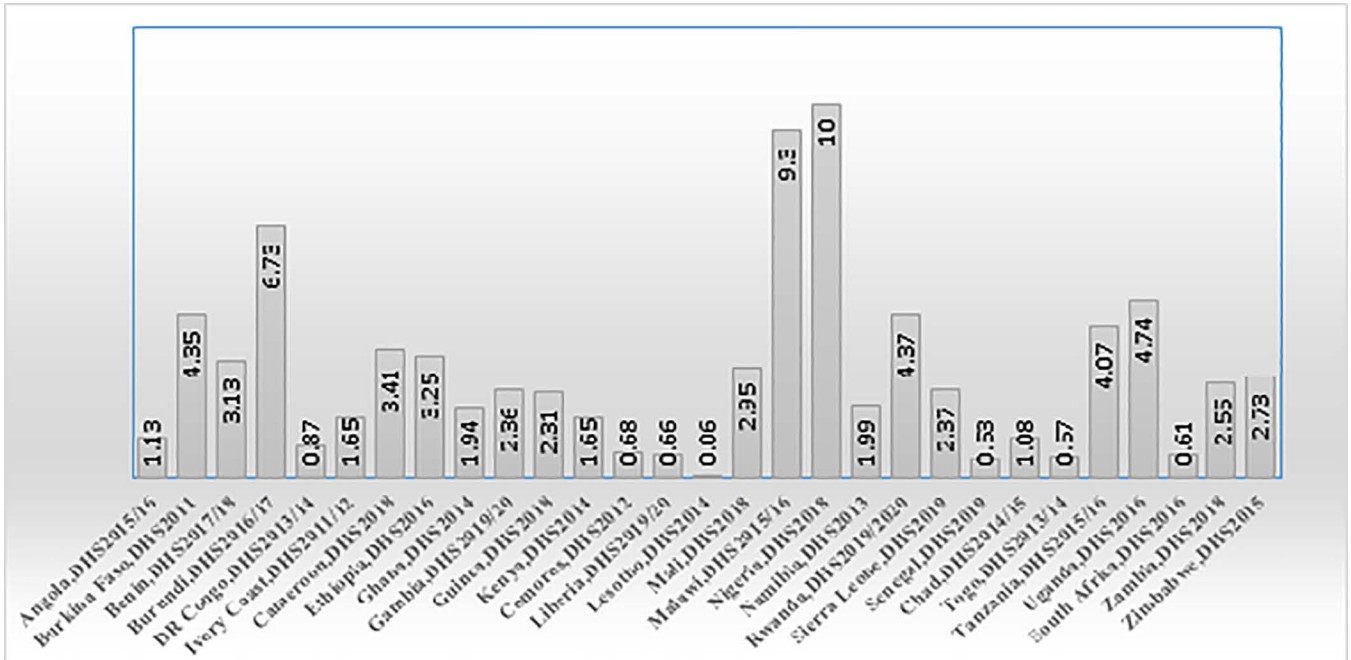

**Fig 1. Proportion of households' solid fuel use in Sub-Saharan African countries.**

In addition, households in the poor and middle wealth categories were 12.46 (AOR = 12.46, 95% CI: 11.34, 13.70) and 4.74 (AOR= 4.74, 95%CI: 4.42, 5.08) times higher odds of solid fuels use compared to rich households, respectively. This disparity highlights the financial barriers poorer households face in accessing cleaner energy sources after controlling other variables.

Households who do not watch television were 2.38 more likely to solid fuel use (AOR = 2.38, 95% CI: 2.23, 2.53) compared to those households watching television.

Households without access to electricity had 3.97 higher odds of solid fuel use (AOR = 3.97, 95% CI: 3.74, 4.22) which emphasizes the importance of electrification in reducing solid fuel dependency.

Larger households with four to seven and more than seven members were (AOR = 3.94, 95% CI: 3.68, 4.21) and 1.92 (AOR= 1.92, 95%CI: 1.84, 2.00) times more likely in solid fuels use compared to those with fewer than four members, respectively. This indicated that larger households are more likely to use solid fuels because of the increased energy demand for cooking and heating, which may lead them to rely on cheaper.

Communities with lower education levels had nearly 2 times higher odds of solid fuel use (AOR = 2.18, 95% CI: 1.84, 2.59) compared to communities with higher education levels. Communities with lower education levels may have limited awareness about the health and environmental consequences of solid fuel use or fewer opportunities to access cleaner energy alternatives.

Similarly, communities with low media exposure had 1.39 times greater odds of solid fuel use (AOR = 1.39, 95% CI: 1.30, 1.49) compared to those with media exposure. Media exposure often provides education and awareness about modern energy sources, government programs, or subsidies promoting clean energy use.

Communities with low and lower-middle-income levels were 63.42 (AOR = 63.42, 95% CI: 56.15, 71.62) and 23.30 (AOR= 23.30, 95%CI: 20.83, 26.07) times higher odds of solid fuel use compared to communities with upper-middle income, respectively. These communities might lack the financial resources to afford cleaner fuels like electricity or lique-fied petroleum gas (LPG). This leads them to rely on readily available and affordable solid fuels such as firewood, char-coal, and crop residues. This high AOR for community-level poverty by exploring the potential lower frequency value of lower middle-income communities than the number of low-income.

Rural households had 3.88 times higher odds of using solid fuel (AOR = 3.88, 95% CI: 3.68, 4.09) compared to urban households, which indicated urban-rural disparities leading to energy infrastructure and access to cleaner fuels. On the other hand, solid fuel use is significantly higher in West Africa (3.50, 3.11–3.94), followed by East Africa (2.21, 1.95–2.49) and Central Africa (1.45, 1.31–1.61), compared to Southern Africa. This disparity in solid fuel use could be attributed to regional differences in socioeconomic development, access to cleaner energy sources, cultural cooking practices, and government policies promoting alternative fuels (Table 2).

## Measures of variations and Model fit statistics

The tailored evidence from AIC, BIC, and Deviance revealed that the final model was the best fit (Table 3). An AUC value closer to 1 suggests that the model has good discriminatory power, while a lower value indicates poorer performance. The area under the ROC curve compares the performance of three models in a fuel type classification, with Model 3 achieving the highest AUC (0.943), indicating the best discriminative ability among the implemented models. Model 1 follows with an AUC of 0.9001, while Model 2 has the lowest (0.8883) value of AUC (Fig 2).

The random intercept variances were 2.94, 2.01, 2.44, and 1.94 for Models 0, I, II, and III, respectively. The null model identified the highest variance, indicating significant differences were observed between groups before adjusting for any predictor variables. While 2.01 was a variability intercept between clusters after considering individual-level predictors. This result revealed that some of the variability between clusters is explained by the individual-level variables. Model II (2.44) showed that the variance between clusters intercepts when only community-level predictors are incorporated. In the last model, where individual and community level factors were included, the lowest (1.94) variance between the cluster

**Table 2. Multilevel logistic regression analysis of individual and community-level factors.**

| Variables | Model 0 (Null model) | Model 1 AOR (95% CI) | Model 2 AOR (95% CI) | Model 3 AOR (95% CI) |
|---|---|---|---|---|
| **Individual level Factors** | | | | |
| Sex of HHH | | | | |
| Male | | 1.01(0.98,1.05) | | 0.78(0.30,1.13) |
| Female | | 1 | | 1 |
| Age of HHH | | | | |
| <35 | | 1 | | 1 |
| 35_60 | | 0.93(0.90,1.16) | | 0.95(0.92,1.99) |
| >60 | | 1.35(1.29,1.41)** | | 1.12(1.05,1.19)** |
| Marital status of HHH | | | | |
| Unmarried | | 1 | | 1 |
| Married | | 1.81(1.75,1.88)** | | 1.14(1.09, 1.20)** |
| Educational status of HHH | | | | |
| No education | | 5.95(5.64,6.28)** | | 4.91(4.59,5.25)** |
| Primary | | 3.71(3.54,3.89)** | | 3.96(3.74,4.20)** |
| Secondary | | 1.37(1.31,1.42)** | | 1.14(1.09,1.20)** |
| Higher | | 1 | | 1 |
| Wealth index | | | | |
| Poor | | 2.37(2.25,2.50)** | | 12.46 (11.34, 13.70)** |
| Middle | | 1.81(1.73,1.89)** | | 4.74(4.42,5.08)** |
| Rich | | 1 | | 1 |
| Watching television | | | | |
| No | | 5.29(5.0,5.50)** | | 2.38(2.23,2.53)** |
| Yes | | 1 | | 1 |
| Accessibility of Electricity | | | | |
| Yes | | 1 | | 1 |
| No | | 5.35(5.10,5.61)** | | 3.97(3.74,4.22)** |
| Number of family | | | | |
| <4 | | 1 | | 1 |
| 4-7 | | 1.91(1.85,1.98)** | | 1.92(1.84,2.00)** |
| >7 | | 3.48(3.31,3.67)** | | 3.94(3.68,4.21)** |
| **Community level factors** | | | | |
| Community level education | | | | |
| Higher | | | 1 | 1 |
| Low | | | 2.04(1.69, 2.45)** | 2.18,1.84,2.59)** |
| Community-level media exposure | | | | |
| Exposed | | | 1 | 1 |
| Unexposed | | | 4.31(4.12, 4.51)** | 1.39(1.30,1.49)** |
| Community level poverty | | | | |
| Low income | | | 44.80(42.62,47.09)** | 63.42(56.15, 71.62)** |
| Lower middle income | | | 14.07(13.400,14.79)** | 23.30(20.83, 26.07)** |
| Upper-middle income | | | 1 | 1 |

*(Continued)*

**Table 2.** (Continued)

| Variables | Model 0 (Null model) | Model 1 AOR (95% CI) | Model 2 AOR (95% CI) | Model 3 AOR (95% CI) |
|---|---|---|---|---|
| Type of residence | | | | |
| Urban | | | 1 | 1 |
| Rural | | | 17.32(16.59,18.08)** | 3.88(3.68,4.09)** |
| Region | | | | |
| East Africa | | | 32.51(31.16,33.92)** | 2.21(1.95, 2.49)** |
| Central Africa | | | 5.22(4.98, 5.46)** | 1.45(1.31,1.61)** |
| West Africa | | | 24.30(23.28,25.37)** | 3.50(3.11,3.94)** |
| Southern Africa | | | 1 | 1 |

Where: 1= reference, **P-value < 0.001(Adjusted OR), *P-value < 0.05(Adjusted OR), HHH =household head, HH=household, Model 0 (Null model) fitted without predictor variables, Model I adjusted for individual-level variables, Model II adjusted for community-level variables, Model III the final model adjusted for both individual- and community-level predictors.

intercepts was observed. The more variables incorporated means most of the variation of the clusters are going to be explained, whereas the remaining variability not explained by the model is decreased. Therefore, the last model is the best to consider variability among clusters.

Since the ICC was significant in an empty model, the authors consider neighborhood variability that would appear in the next more complex models. The ICC value of 0.47 in the null model showed that 47% proportion of the total variance in solid fuel use is due to differences between clusters. The lower ICC value (0.37) from the fourth model indicated that it explained some (37%) of the between-cluster differences in solid fuel use. This lower ICC in the final model unmeasured variables, such as cultural factors, household income, or access to cleaner energy alternatives, which could contribute to the unexplained variance in solid fuel. The MOR is used to quantify the variation between clusters by comparing two persons from two randomly selected clusters. The second model showed that the MOR was 4.44. That means the household found in the higher solid fuel use cluster was 4.44 times higher than the household found in another cluster with lower solid fuel use, but these two households have identical covariates. This indicated that the source of difference is the cluster difference where they are located.

## Discussion

This study used data from 29 DHS surveys to find out how fixed and random influences affected the use of solid fuels in Sub-Saharan Africa. The prevalence of solid fuel use in sub-Saharan Africa was 82.05%, with 95%CI (81.90, 82.21). This finding is higher than the finding of the research done in India, where 40% of households still rely on solid fuels [28]. This result is lower than the prevalence of solid fuel use in Ghana, 85% [29]. On the other hand, this finding is higher than the result of the study conducted internationally (62%) among 150 countries [30]. These differences could be due to sample size variation, study period, and setting. On the other hand, this finding is consistent with the study conducted in Bangladesh (82%) [31].

The use of solid fuels varies significantly throughout Sub-Saharan Africa at the national and local levels, according to the random effects study. The variance estimates, which vary from 1.94 to 2.94, show that contextual factors have a significant impact on household energy decisions. Living in different communities or nations considerably changes the likelihood of using solid fuels, as seen by the Median Odds Ratio (MOR), which ranges from 3.78 to 5.13. This further emphasizes the significant importance of contextual factors that cannot be assessed. Furthermore, the proportional change in variance (PCV) highlights the significance of contextual variables like infrastructure and policy by showing that the inclusion of fixed effects can account for up to 93% of the variance at higher levels. Last but not least, the Intraclass Correlation Coefficient

(ICC) values indicate that a significant amount of the variability—between 37% and 47%—is due to variations between communities and nations, indicating the necessity of multi-level and localized policy interventions to successfully address solid fuel dependency. This implies that this study reveals that household solid fuel use in Sub-Saharan Africa is significantly influenced by contextual, cultural, and structural factors at national and local levels, emphasizing the need for policy interventions at East Africa, Central Africa, and West Africa to address these variations effectively.

Based on the final model included individual-level variables such as the household head's age, sex, marital status, level of education, wealth index, number of family members, and television watching, as well as community-level variables such as the number of family members, accessibility to electricity, media exposure, and type of residence that were statistically significant in determining solid fuel use.

**Table 3. Measures of variations and Model fit statistics of included models.**

| Measures of variations in random effect | | | | |
|---|---|---|---|---|
| Metrics | Model 0 (Null model) | Model 1 AOR (95% CI) | Model 2 AOR (95% CI) | Model 3 AOR (95% CI) |
| Variance | 2.94 | 2.01 | 2.44 | 1.94 |
| MOR | 5.13 | 3.87 | 4.44 | 3.78 |
| PCV | 1 | 93.00% | 50.00% | 50.00% |
| ICC | 0.47(0.45,0.49) | 0.38(0.36,0.40) | 0.43(0.40,0.45) | 0.37(0.35,0.40) |
| **Model fitness test statistics** | | | | |
| AIC | 207,562.10 | 126,115.50 | 113,873.30 | 84,644.42 |
| BIC | 207,582.80 | 126,270.50 | 113,945.30 | 84,849.47 |
| Deviance | 207,558.06 | 126,085.50 | 113,859.34 | 84,604.42 |

Where: AIC=Akaike's information criteria, BIC=Bayesian information criteria, ICC = intraclass correlation, MOR = median odds ratio, PCV= proportional change in variance

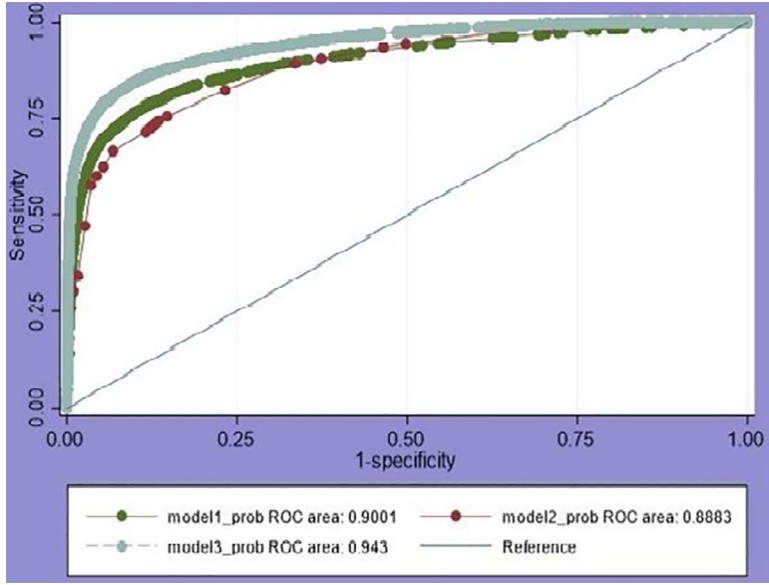

**Fig 2. Area under the ROC curve in accuracy measurement across the three models.**

Households headed by individuals aged >60 years have higher odds of using solid fuel compared to those households headed by younger counterparts. This might redirect an age difference in energy preferences or access, where older household heads could be more familiar with traditional fuel sources or less likely to adopt newer technologies like electricity or gas for cooking. This finding was supported by previous studies done in different parts of the world [19,28,32] but contradicted other studies done in Cameroon [33].

Married household heads are associated with a greater likelihood of using solid fuels than unmarried heads. This could be due to larger household sizes typically found in married households, which might require more energy for cooking and heating, thus high reliance on solid fuels use. This finding is consistent with the studies done in Uganda [34] and Ghana [20,32].

Educational status plays a significant role, with households whose heads have no formal education, primary education, and secondary education being more likely to depend on solid fuel use compared to those with higher education. This may indicate that higher education levels correlate with greater awareness of the health effects of cleaner dirty sources of energy, as well as the highest educational level increases financial capacity to afford clear energy source options. This finding is supported by other earlier research works [19,32–36]

The wealth index is the other critical determinant, with poor households being much more likely to rely on solid fuels compared to rich households. This indicated that poor economic status is a major barrier to accessing cleaner energy sources, as poorer households may lack the financial means to invest in alternatives like gas or electricity [20,33,35,36].

Households where the families do not watch television have higher odds of using solid fuels than family members who watch television. The possible explanation is that this could reflect the role of media exposure in raising consciousness about the health risks of solid fuel use and the benefits of cleaner energy choices, leading to a reduced reliance on solid fuels.

Households without access to electricity are more likely to use solid fuels compared to households with access to electricity. This suggests that electricity access is a significantly associated factor in reducing solid fuel use, as households without electricity often have limited choices and must depend on solid fuel use like biomass, which can be easily accessed. This finding is similar to the study conducted in Afghanistan [37].

Households with larger family members have significantly higher odds of using solid fuels compared to smaller family member households. A possible justification is that larger households typically have greater energy demand, which might be more dependent on inexpensive energy sources met with solid fuels than with modern energy sources. This result is consistent with the findings of earlier studies [20,28,35,38].

Communities with lower levels of education are more likely to use solid fuels than communities with higher levels of education. This showed that community awareness and collective knowledge influence energy choices, whereas less educated communities might be less informed about cleaner choices or face cultural inertia in adopting new fuel types. In addition, communities with higher education are expected to be financially rich to lead a safe life. This finding is in line with the results obtained in different parts of the world [19,27,35]

Community-level media exposure was the other determinant of fuel source choice. Communities with low media exposure have higher odds of using solid fuels. This could be due to the media being a powerful tool for spreading information about the problems of solid fuel use and the availability of other energy sources, thereby prompting community-level energy use behavior. This result is supported by earlier research works done in various parts of the world [19,22,23].

Community-level poverty was a statistically significant and pertinent predictor **of** energy fuel sources in sub-Saharan Africa. Communities with lower income levels revealed extremely higher odds of using solid fuels compared to higher-income communities. This finding could be due to the broader socio-economic, environmental, and policy contexts, considering the broader structural factors that contribute to poverty and fuel choices. This emphasizes the broader economic barriers to cleaner energy access within impoverished communities, where limited financial resources restrict the transition away from traditional biomass. This finding is in line with other studies conducted in Ethiopia [19,22,23].

Rural households are significantly more likely to use solid fuels compared to urban households. This disparity reflects infrastructural challenges in rural areas, such as limited access to electricity and clean cooking options, along with a cultural inclination toward using locally available and adaptable biomass fuel. This result is supported by the study conducted in Ethiopia [19,22,23], Ghana [39], and Vietnam [40]. The study's findings advised that governments should provide electricity, particularly to rural and urban poor citizens who cannot afford to use renewable energy. Furthermore, it is vital to raise awareness of the potential health risks connected with solid fuel consumption among households with low educational attainment and media exposure. The current study did not include all possible factors of fuel selection of households. In addition, the findings of the study were from secondary DHS data.

The results show significant regional differences in the usage of solid fuels throughout Sub-Saharan Africa, with West Africa having the highest incidence compared to Southern Africa, followed by East and Central Africa. This pattern might result from differences in these regions' socioeconomic development, availability of cleaner energy sources, cultural customs, and legislative initiatives. Lower rates of electrification, a lack of reasonably priced alternatives, and a long-standing dependence on traditional cooking techniques may all be contributing factors to West Africa's noticeably greater reliance on solid fuels. These differences demonstrate the necessity of region-specific initiatives to encourage energy transitions and lessen the negative effects that using solid fuels has on the environment and human health.

Community-level issues like poverty and education can influence individual decisions by influencing access to resources, social norms, and opportunities. These decisions, in turn, can affect individual actions and results. Higher educational attainment within a community may encourage better decision-making and healthier behaviors, whereas those in disadvantaged communities may have restricted access to healthcare or education, which could result in worse health outcomes. All of these elements work together to affect people's behavior and support the more general trends in social, economic, and health outcomes.

The use of secondary data from DHS has several limitations that need to be carefully considered. First, because the researchers have no control over the data collecting procedure or sampling techniques, using pre-existing datasets may create selection bias. Second, the lack of control or customization over variable selection may restrict the analysis's breadth and cause potentially significant elements to be overlooked. Third, reporting biases, such as recall and social desirability biases, could affect the data, particularly if it is self-reported. The fourth comparison of the prevalence was with specific countries. Finally, yet importantly, because DHS data is cross-sectional and only records information at one point in time rather than over an extended period, it limits the capacity to draw causal links. To address the above limitations of the current study, future studies should consider exploring longitudinal data to establish causal relationships and incorporating qualitative methods to gain deeper insights into household decision-making processes related to solid fuel use.

The determinants of solid fuel use in Sub-Saharan Africa, as identified through multilevel analysis using DHS data, have significant implications for local policy and practice development. Key factors such as socioeconomic status, education levels, rural-urban disparities, and access to cleaner energy alternatives highlight the need for targeted interventions. Policymakers should prioritize improving access to affordable and sustainable energy solutions, particularly in rural and low-income areas, while integrating energy access with broader development goals such as education, poverty alleviation, and healthcare. Local practices should focus on community-based awareness campaigns to promote the benefits of cleaner fuels and efficient cooking technologies, alongside subsidies or incentives to reduce the financial barriers to adoption. Additionally, addressing gender disparities, as women are disproportionately affected by solid fuel use, should be central to policy design, ensuring that interventions are inclusive and equitable. Collaborative efforts between governments, NGOs, and private sectors are essential to create sustainable, long-term solutions that align with global health and environmental targets.

## Conclusion

The study showed that the prevalence of solid fuel use was high in sub-Saharan Africa. Factors such as the age of the household head and marital status of the household head, educational status of the household head, wealth index,

watching television, access to electricity, family size, community-level education, community-level media exposure, residence, income level, and community-level poverty were significantly associated with solid fuel use. Solid fuel cooking has been related to respiratory and cardiovascular problems such as lung cancer, chronic obstructive pulmonary disease, and heart disease, as well as pneumonia in children, the elderly, and women who spend most of their time at home. The collected wood further causes countries' deforestation. Therefore, health policymakers, NGOs, and other possible stakeholders need to work on increasing access to electricity, expanding education, increasing media exposure, reducing poverty, and reducing urban-rural disparities and solid fuel use. Customized programs concentrating on socioeconomic development, increased access to cleaner energy, and cultural and legal adaptation are essential for fostering energy transitions and reducing the negative effects on the environment and human health to address the regional differences in solid fuel consumption throughout Sub-Saharan Africa.

## Acknowledgments

The authors would like to thank the Demographic and Health Survey (DHS) Program for permitting us to use the Ethiopian DHS dataset.

## Author contributions

**Conceptualization:** Jember Azanaw, Mastewal Endalew.

**Data curation:** Jember Azanaw, Mastewal Endalew.

**Formal analysis:** Jember Azanaw.

**Investigation:** Jember Azanaw, Mastewal Endalew.

**Methodology:** Jember Azanaw, Mastewal Endalew.

**Project administration:** Jember Azanaw.

**Resources:** Jember Azanaw.

**Software:** Jember Azanaw.

**Supervision:** Jember Azanaw.

**Validation:** Jember Azanaw, Mastewal Endalew.

**Visualization:** Jember Azanaw, Mastewal Endalew.

**Writing – original draft:** Jember Azanaw, Mastewal Endalew.

**Writing – review & editing:** Jember Azanaw, Mastewal Endalew.

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
