## [Decision Letter · Decision Letter 0]

10 Jan 2025

PONE-D-24-55055Analyzing Fixed and Random Effects on Solid Fuel Use in Sub-Saharan Africa: Multilevel Mixed-Effects Modeling Approach with 29 DHS DataPLOS ONE

Dear Dr. Azanaw,

Thank you for submitting your manuscript to PLOS ONE. After careful consideration, we feel that it has merit but does not fully meet PLOS ONE’s publication criteria as it currently stands. Therefore, we invite you to submit a revised version of the manuscript that addresses the points raised during the review process.

Both reviewers have raised a number of points that require careful attention and changes to the manuscript.

We look forward to receiving your revised manuscript.

Kind regards,

Alison Parker

Academic Editor

PLOS ONE

Journal Requirements:

2. In the online submission form, you indicated that your data is available only on request from a third party. Please note that your Data Availability Statement is currently missing the contact details for the third party, such as an email address or a link to where data requests can be made. Please update your statement with the missing information.

Reviewers' comments:

Reviewer's Responses to Questions

**Comments to the Author**

1. Is the manuscript technically sound, and do the data support the conclusions?

Reviewer #1: No

Reviewer #2: Partly

2. Has the statistical analysis been performed appropriately and rigorously? 

Reviewer #1: Yes

Reviewer #2: No

3. Have the authors made all data underlying the findings in their manuscript fully available?

Reviewer #1: Yes

Reviewer #2: No

4. Is the manuscript presented in an intelligible fashion and written in standard English?

Reviewer #1: No

Reviewer #2: No

5. Review Comments to the Author

Reviewer #1: General Comment:

The study addresses an important public health and environmental issue- solid fuel in Sub-Saharan Africa. The methodological approach is appropriate to the nature of the data and uses multilevel mixed-effects modeling, taking into consideration the hierarchical structure of the data. The dataset included 29 DHS surveys for comprehensive regional coverage, hence a large sample size of 233,391 households.

However, the manuscript requires thorough proofreading. Poor grammar and awkward phrasing significantly detract from the content of the manuscript; for example, "Since the human being exists on land.". The findings lack a strong connection with the existing literature. Comparisons are made with other regions, but implications for local policy and practice development are not well developed. The inclusion of P-values along with the AOR is unclear in some sections and needs further explanation. The abstract has failed to provide a clearly articulate the research question and the novelty of this study. The term "Mixed and Random Effects" in the title is ambiguous and might be puzzling for some readers. Whereas it was able to identify factors influencing solid fuel use, it has not indicated how the findings could provide useful information for specific interventions or policies.

Detailed Comments:

Title:

01. The title could be simplified for clarity, e.g., “Determinants of Solid Fuel Use in Sub-Saharan Africa: A Multilevel Analysis Using DHS Data.”

02. The term "Mixed and Random Effects" in the title is ambiguous and might be puzzling for some readers.

Abstract:

03. The first sentence of the abstract is unnecessarily broad.

04. The abstract has failed to clearly articulate this study's research question and novelty.

05. The aim of the study written in the abstract is not clear.

06. Emphasize actionable insights and policy recommendations in the conclusion of the abstract.

Introduction:

07. The introduction highlights the problem very well but does not provide a clear articulation of the research gap. For instance, while it mentions previous studies on small samples or in specific regions, how this larger dataset adds value needs to be better stated. Citations are irregularly formatted and need to follow the journal's formatting rules.

Methods:

08. The multilevel modeling approach has been appropriately described, though model selection and handling of missing data need further clarity.

09. On ethics approval, it says, "Not applicable" since the data used are secondary; however, explaining in detail, how the DHS protects anonymity and adheres to the ethical requirements, would strengthen this section.

10. The size and scope of the data used in the manuscript require an explanation of how missing data were treated.

11. Although many variables are represented, there is little explanation of why certain predictors were chosen and how they relate to the existing theories or frameworks on solid fuel use. The operationalization of the variables, particularly those at the community level, is not fully described about poverty and education levels.

Results:

12. The prevalence of the use of solid fuel is presented quite clearly; tables and figures should be better formatted for readability.

13. Results must integrate findings into contextual interpretations and not merely state statistical outputs. For instance, the discussion of the Adjusted Odds Ratios (e.g., AOR = 12.46 for poor households) could include real-world implications.

14. Table heads and figure captions should be intelligible independently.

15. The magnitude of the odds ratios seems plausible, but the authors could interpret the outliers or very high AOR values-for example, AOR = 144.49 for community-level poverty-more critically. Such extreme values are indicative of highly stratified populations or possibly overlapping variables.

16. The AUC is well described but could be briefly explained for readers who may not be familiar with this measure.

17. The odds ratio for low-income communities seems very high (AOR = 144.49), questionable about the granularity of the poverty measure. The author should check the causes.

18. The study has shown that even the most complete model- Model III-explains only a small proportion of the variance ICC = 0.37. the authors should present possible unmeasured variables.

Discussion:

19. While the discussion mentions broad recommendations, it lacks specificity. The authors miss an opportunity to provide evidence-based, region-specific recommendations for policymakers.

20. While the discussion mentions broad recommendations, it lacks specificity.

21. While the authors recognize the structural and cultural barriers, actionable recommendations-such as specific programs or policies aimed at reducing dependence on solid fuels are in need of more emphasis in this section.

22. Authors should critically review the limitations regarding the use of secondary data from DHS, including possible biases and inability to control the selection of variables.

23. The reported prevalence of solid fuel use is identified and described in the result section of this study which befits the large dataset that this study leveraged. Nevertheless, the presentation might have been even more contextual, for instance, comparing it with certain countries outside of Sub-Saharan Africa or global averages.

24. While household heads of older age were more likely to use solid fuel, it was specifically with the categories of age >60 years, AOR = 1.12. The study speculates on generational differences in energy preferences; however, the discussion fails to provide empirical evidence or citations to support this claim.

25. The authors miss an opportunity to provide evidence-based, region-specific recommendations for policymakers.

26. Through the discussion mentioned the findings of other studies however it misses the comparison or contrast of findings.

27. Whereas the authors have focused on individual-level predictors, community-level factors i.e., poverty and education remain underexplored in the discussion. The discussion may be further strengthened by exploring the dynamics of how community-level factors shape or influence individual decisions.

28. While limitations are recognized, they are only superficially discussed. The study does not discuss the possible bias from using self-reported data or the limitations of secondary data. The greatest limitation is the cross-sectional nature of the analysis, which should be discussed more critically. The longitudinal data might have shown causality, which is missing here.

Conclusion:

29. The conclusion summarizes the findings appropriately but should highlight the study's novelty and its practical implications, such as recommendations for policymakers.

Overall comment:

The manuscript needs a lot of work in terms of writing quality, contextualization of findings, and depth of discussion.

Reviewer #2: Following are my comments regarding the paper:

1) The title of the paper seems misleading, as the primary focus appears to be on the determinants of solid fuel usage in sub-Saharan Africa (SSA). The title should better reflect the main objectives and scope of the study.

2) The introduction/background and contribution/value addition of that study fail to generate significant interest. The authors should present more compelling points to address these problems.

3) The paper is missing a literature review section. The authors should discuss prior studies in the field, identify gaps in the existing literature, and justify how their approach addresses these gaps. This will provide a stronger foundation for the research.

4) The rationale for using the multilevel logistic regression model is not sufficiently explained. The authors should provide a thorough justification for why this model is appropriate for the study, supported by relevant literature.

5) The Demographic and Health Surveys (DHS) were conducted in different years for various countries in SSA. The authors should clearly explain how they combined data from different households and years into a single dataset. They must address potential issues that may arise from merging datasets of varying sizes and discuss why this approach does not compromise the validity of the analysis.

6) The authors state, “Variables with a P-value <.25 in the bivariate analysis were selected for multivariate analysis.” This threshold requires further explanation and justification, as it is not a standard practice in most studies.

7) The study primarily relies on a multilevel logistic regression model. It is essential to perform robustness checks using at least one alternative, more advanced statistical model to validate the baseline findings and ensure their reliability.

8) The policy implications section should be expanded, offering detailed and actionable recommendations based on the findings. Additionally, the paper should discuss the study’s limitations and suggest directions for future research to address any unresolved issues.

6. PLOS authors have the option to publish the peer review history of their article (what does this mean? ). If published, this will include your full peer review and any attached files.

**Do you want your identity to be public for this peer review?** For information about this choice, including consent withdrawal, please see our Privacy Policy .

Reviewer #1: No

Reviewer #2: **Yes: ** Md Abdullah Omar

---

## [Author Response · Author response to Decision Letter 1]

30 Jan 2025

Point-by-point response to editor and reviewers

Please accept our revised manuscript and note our point-by point response to reviewer below for the manuscript titled " PONE-D-24-55055, Analyzing Fixed and Random Effects on Solid Fuel Use in Sub-Saharan Africa: Multilevel Mixed-Effects Modeling Approach with 29 DHS Data

PLOS ONE "

Note: Responses for each issue raised by Academic Editor and Reviewers indicated by highlight.

Authors’ response: Thank you dear editor, we have seen all your concerns and done as per you recommendation.

Reviewer Comments:

Reviewer #1

Reviewer #1: General Comment:

The study addresses an important public health and environmental issue- solid fuel in Sub-Saharan Africa. The methodological approach is appropriate to the nature of the data and uses multilevel mixed-effects modeling, taking into consideration the hierarchical structure of the data. The dataset included 29 DHS surveys for comprehensive regional coverage, hence a large sample size of 233,391 households.

However, the manuscript requires thorough proofreading. Poor grammar and awkward phrasing significantly detract from the content of the manuscript; for example, "Since the human being exists on land.". The findings lack a strong connection with the existing literature. Comparisons are made with other regions, but implications for local policy and practice development are not well developed. The inclusion of P-values along with the AOR is unclear in some sections and needs further explanation. The abstract has failed to provide a clearly articulate the research question and the novelty of this study. The term "Mixed and Random Effects" in the title is ambiguous and might be puzzling for some readers. Whereas it was able to identify factors influencing solid fuel use, it has not indicated how the findings could provide useful information for specific interventions or policies.

Authors’ response: Thank you very much dear reviewer for your time and effort for showing authors gaps and for constructive comments to be corrected. Based on these comments authors addressed all issues raised.

Detailed Comments:

Title:

01. The title could be simplified for clarity, e.g., “Determinants of Solid Fuel Use in Sub-Saharan Africa: A Multilevel Analysis Using DHS Data.”

Authors’ response: we thank for this comment and corrected as recommended.

02. The term "Mixed and Random Effects" in the title is ambiguous and might be puzzling for some readers.

Authors’ response: Sorry for such inconvenient but now corrected.

Abstract:

03. The first sentence of the abstract is unnecessarily broad.

Authors’ response: Authors thank for this comment and we have rephrased this sentence.

04. The abstract has failed to clearly articulate this study's research question and novelty.

Authors’ response: Thank you very much dear reviewer, authors rephrased this section.

05. The aim of the study written in the abstract is not clear.

Authors’ response: Authors thank you the reviewer and we rephrased.

06. Emphasize actionable insights and policy recommendations in the conclusion of the abstract.

Authors’ response: Authors appreciate the reviewer issues and we corrected as per your recommendation.

Introduction:

07. The introduction highlights the problem very well but does not provide a clear articulation of the research gap. For instance, while it mentions previous studies on small samples or in specific regions, how this larger dataset adds value needs to be better stated. Citations are irregularly formatted and need to follow the journal's formatting rules.

Authors’ response: we thank you for the very valuable comments in this regard and authors as corrected as suggested by the reviewer.

Methods:

08. The multilevel modeling approach has been appropriately described, though model selection and handling of missing data need further clarity.

Authors’ response: Thank you very much dear reviewer for this comment. We revised this section.

09. On ethics approval, it says, "Not applicable" since the data used are secondary; however, explaining in detail, how the DHS protects anonymity and adheres to the ethical requirements would strengthen this section.

Authors’ response: Thank you very much for the comment. We accept the comment and the authors included the reviewer's idea.

10. The size and scope of the data used in the manuscript require an explanation of how missing data were treated.

Authors’ response: Thank you very much dear reviewer for this comment. We revised this section.

11. Although many variables are represented, there is little explanation of why certain predictors were chosen and how they relate to the existing theories or frameworks on solid fuel use. The operationalization of the variables, particularly those at the community level, is not fully described about poverty and education levels.

Authors’ response: Thank you very much for the issues. The proportion of households is categorized as low poverty level (those with ≥ 50%) and higher poverty level (those with <50%) using the national median value. As well as community-level education, the same way as the poverty level is defined. Therefore, we included it in the manuscript.

Results:

12. The prevalence of the use of solid fuel is presented quite clearly; tables and figures should be better formatted for readability.

Author’s response: We, authors would really thank for pointing us such very relevant comments and these were paraphrased.

13. Results must integrate findings into contextual interpretations and not merely state statistical outputs. For instance, the discussion of the Adjusted Odds Ratios (e.g., AOR = 12.46 for poor households) could include real-world implications.

Authors’ response: Thank you very much for the comment. Corrected as suggested by the reviewer.

14. Table heads and figure captions should be intelligible independently.

Authors’ Response: Thank you for your insightful comment. We have revised the table heads and figure captions to ensure they are self-explanatory.

15. The magnitude of the odds ratios seems plausible, but the authors could interpret the outliers or very high AOR values-for example, AOR = 144.49 for community-level poverty-more critically. Such extreme values are indicative of highly stratified populations or possibly overlapping variables.

Authors’ Response: Thank you for your valuable feedback and authors’ include possible reason.

16. The AUC is well described but could be briefly explained for readers who may not be familiar with this measure.

Authors’ response: we thank for this appreciated comment to these issues and we modified as per suggested by the reviewer.

17. The odds ratio for low-income communities seems very high (AOR = 144.49), questionable about the granularity of the poverty measure. The author should check the causes.

Authors’ Response: Thank you for your valuable feedback and authors’ include possible cause of for such high AOR.

18. The study has shown that even the most complete model- Model III-explains only a small proportion of the variance ICC = 0.37. the authors should present possible unmeasured variables.

Authors’ Response: Thank you for your insightful comment and authors included possible unmeasured variables.

Discussion:

19. While the discussion mentions broad recommendations, it lacks specificity. The authors miss an opportunity to provide evidence-based, region-specific recommendations for policymakers.

Authors’ Response: Thank you for your valuable feedback. We believe this will strengthen the overall impact of our work and provide actionable insights for local decision-makers. Therefore, the authors include this pertinent factor.

20. While the discussion mentions broad recommendations, it lacks specificity.

Authors’ Response: We appreciate the reviewer’s feedback and we enhanced the discussion by including specific, actionable recommendations tailored to the study findings to provide greater clarity and practical value.

21. While the authors recognize the structural and cultural barriers, actionable recommendations-such as specific programs or policies aimed at reducing dependence on solid fuels are in need of more emphasis in this section.

Authors’ Response: We have incorporated these recommendations in the revised section to ensure actionable and context-specific solutions are adequately emphasized. We trust this enhancement strengthens the manuscript and aligns with your suggestions.

22. Authors should critically review the limitations regarding the use of secondary data from DHS, including possible biases and inability to control the selection of variables.

Authors’ Response: Thank you for the insightful comment. We acknowledge that using secondary data from the Demographic and Health Surveys (DHS) has inherent limitations, which we have now addressed more explicitly in the revised manuscript.

23. The reported prevalence of solid fuel use is identified and described in the result section of this study which befits the large dataset that this study leveraged. Nevertheless, the presentation might have been contextual, for instance, comparing it with certain countries outside of Sub-Saharan Africa or global averages.

Authors’ Response: Thank you for your thoughtful suggestion. We appreciate your emphasis on contextualizing the reported prevalence of solid fuel use. While the current study primarily focuses on trends and determinants of solid fuel use within Sub-Saharan Africa (SSA), we agree that comparing these findings with countries outside SSA or global averages would provide additional valuable context. We believe this addition will enhance the contextual value of the study and appreciate your valuable input in improving our work.

24. While household heads of older age were more likely to use solid fuel, it was specifically with the categories of age >60 years, AOR = 1.12. The study speculates on generational differences in energy preferences; however, the discussion fails to provide empirical evidence or citations to support this claim.

Authors’ Response: Thank you for this interesting comment we included this revision this revision to address the reviewer’s concerns.

25. The authors miss an opportunity to provide evidence-based, region-specific recommendations for policymakers.

Authors’ Response: we thank you dear reviewer, accept the suggestion, and included region-specific, evidence-based recommendations for policymakers in the revised manuscript.

26. Through the discussion mentioned the findings of other studies however, it misses the comparison or contrast of findings.

Authors’ response: Authors thank you dear reviewer, we have compared it with previous studies even if it includes are country, but put as a limitation

27. Whereas the authors have focused on individual-level predictors, community-level factors i.e., poverty and education remain underexplored in the discussion. The discussion may be further strengthened by exploring the dynamics of how community-level factors shape or influence individual decisions.

Authors’ response: Authors thank you, dear reviewer; we have included strengthening the manuscript.

28. While limitations are recognized, they are only superficially discussed. The study does not discuss the possible bias from using self-reported data or the limitations of secondary data. The greatest limitation is the cross-sectional nature of the analysis, which should be discussed more critically. The longitudinal data might have shown causality, which is missing here.

Authors’ Response: Thank you dear reviewer, we have incorporated these limitations in the revised manuscript.

Conclusion:

29. The conclusion summarizes the findings appropriately but should highlight the study's novelty and its practical implications, such as recommendations for policymakerss.

Authors’ Response: we thank you, dear reviewer, accept the suggestion, and include region-specific, evidence-based recommendations for policymakers in the revised manuscript.

Overall comment:

The manuscript needs a lot of work in terms of writing quality, contextualization of findings, and depth of discussion.

Authors’ Response: Thank you for your feedback; I will revise the manuscript to improve the writing quality, better contextualize the findings, and deepen the discussion.

Reviewer #2: Following are my comments regarding the paper:

1) The title of the paper seems misleading, as the primary focus appears to be on the determinants of solid fuel usage in sub-Saharan Africa (SSA). The title should better reflect the main objectives and scope of the study.

Authors’ Response: We appreciate the reviewer's comment and have revised the title to more accurately reflect the study's primary focus on the determinants of solid fuel usage in Sub-Saharan Africa.

2) The introduction/background and contribution/value addition of that study fail to generate significant interest. The authors should present more compelling points to address these problems.

Authors’ Response: Thank you for your comment and we have revised the introduction to emphasize the significance of the study by clearly outlining the research gaps, highlighting its innovative approach, and demonstrating its potential impact in addressing the identified problems.

3) The paper is missing a literature review section. The authors should discuss prior studies in the field, identify gaps in the existing literature, and justify how their approach addresses these gaps. This will provide a stronger foundation for the research.

Authors’ Response: We sincerely thank the reviewer for their valuable feedback and insightful comments, which have significantly improved the quality and clarity of our manuscript. So we have revised this part.

4) The rationale for using the multilevel logistic regression model is not sufficiently explained. The authors should provide a thorough justification for why this model is appropriate for the study, supported by relevant literature.

Authors’ Response: Thank you for this valuable comment and we have clarified this in the revised manuscript.

5) The Demographic and Health Surveys (DHS) were conducted in different years for various countries in SSA. The authors should clearly explain how they combined data from different households and years into a single dataset. They must address potential issues that may arise from merging datasets of varying sizes and discuss why this approach does not compromise the validity of the analysis.

Authors’ Response: We appreciate the reviewer's comment and have addressed this concern.

6) The authors state, “Variables with a P-value <.25 in the bivariate analysis were selected for multivariate analysis.” This threshold requires further explanation and justification, as it is not a standard practice in most studies.

Authors’ Response: Thank you for highlighting this point. We have now added relevant references to justify this approach and included a more detailed explanation in the manuscript.

7) The study primarily relies on a multilevel logistic regression model. It is essential to perform robustness checks using at least one alternative, more advanced statistical model to validate the baseline findings and ensure their reliability.

Authors’ Response: We appreciate the reviewer's insightful comment regarding the need for robustness checks using an alternative statistical model. In this study, we primarily relied on a multilevel logistic regression model, as it is well-suited for addressing the hierarchical structure of the data. To ensure the validity of our findings, we conducted rigorous model diagnostics, including the Akaike Information Criterion (AIC), Bayesian Information Criterion (BIC), deviance, and the area under the Receiver Operating Characteristic (ROC) curve, which are widely recognized measures of model performance and fit. These metrics provide robust evidence for the reliability of our results. Moreover, similar studies in the field have consistently relied on such diagnostics to validate their findings without employing alternative advanced models. While we acknowledge the value of additional robustness checks, attempting to incorporate an alternative statistical model would necessitate significant changes to the entire analytical framework, potentially shifting the focus of the stu

---

## [Editor Report · Decision Letter 1]

4 Feb 2025

PONE-D-24-55055R1Determinants of Solid Fuel Use in Sub-Saharan Africa: A Multilevel Analysis Using DHS DataPLOS ONE

Dear Dr. Azanaw,

Thank you for submitting your manuscript to PLOS ONE. After careful consideration, we feel that it has merit but does not fully meet PLOS ONE’s publication criteria as it currently stands. Therefore, we invite you to submit a revised version of the manuscript that addresses the points raised during the review process. Please provide a more detailed response to reviewers, outlining exactly what changes have been made and the line numbers where the changes have bee made.

We look forward to receiving your revised manuscript.

Kind regards,

Alison Parker

Academic Editor

PLOS ONE

---

## [Author Response · Author response to Decision Letter 2]

5 Feb 2025

A point-by-point response to editor and reviewers

Please accept our revised manuscript and note our point-by-point response to the reviewer below for the manuscript titled " PONE-D-24-55055, Analyzing Fixed and Random Effects on Solid Fuel Use in Sub-Saharan Africa: Multilevel Mixed-Effects Modeling Approach with 29 DHS Data

PLOS ONE "

Note: Responses for each issue raised by the Academic Editor and Reviewers are indicated by highlight.

Authors’ response: Thank you, dear editor, we have seen all your concerns and have done as per you recommendation.

Reviewer Comments:

Reviewer #1

Reviewer #1: General Comment:

The study addresses an important public health and environmental issue- solid fuel in Sub-Saharan Africa. The methodological approach is appropriate to the nature of the data and uses multilevel mixed-effects modeling, taking into consideration the hierarchical structure of the data. The dataset included 29 DHS surveys for comprehensive regional coverage, hence a large sample size of 233,391 households.

However, the manuscript requires thorough proofreading. Poor grammar and awkward phrasing significantly detract from the content of the manuscript; for example, "Since the human being exists on land.".

Authors’ response: Thank you very much dear reviewer for your time and effort in showing authors gaps and for constructive comments to be corrected. So authors rephrased from line numbers 25 to 26.

The findings lack a strong connection with the existing literature.

Authors’ response: Thank you very much dear reviewer we have included kinds of literature from line numbers 85 to 91.

Comparisons are made with other regions, but implications for local policy and practice development are not well developed.

Authors’ response: Thank you very much dear reviewer we have included implications for local policy and practice from line numbers 442 to 454.

The inclusion of P-values along with the AOR is unclear in some sections and needs further explanation.

Authors’ response: Thank you very much dear reviewer we have explanations of P-values along with the AOR from line numbers 222 to 286.

The abstract has failed to provide a clearly articulate the research question and the novelty of this study.

Authors’ response: Thank you very much dear reviewer we have included the research question and the novelty from line numbers 30 to 34.

The term "Mixed and Random Effects" in the title is ambiguous and might be puzzling for some readers. Whereas it was able to identify factors influencing solid fuel use, it has not indicated how the findings could provide useful information for specific interventions or policies.

Authors’ response: Thank you very much; dear reviewer authors corrected the title as recommended from line numbers 1 to 2.

Detailed Comments:

Title:

01. The title could be simplified for clarity, e.g., “Determinants of Solid Fuel Use in Sub-Saharan Africa: A Multilevel Analysis Using DHS Data.”

Authors’ response: we thank you for this comment and corrected it as recommended from line numbers 1 to 2.

02. The term "Mixed and Random Effects" in the title is ambiguous and might be puzzling for some readers.

Authors’ response: Sorry for such an inconvenience but now corrected from line numbers 1 to 2.

Abstract:

03. The first sentence of the abstract is unnecessarily broad.

Authors’ response: Authors thank you for this comment and we have rephrased this sentence from line numbers 25 to 26.

04. The abstract has failed to clearly articulate this study's research question and novelty.

Authors’ response: Thank you very much dear reviewer we have included the research question and the novelty from line numbers 30 to 34.

05. The aim of the study written in the abstract is not clear.

Authors’ response: Authors thank you the reviewer and we rephrased to clear from line numbers 33 to 34.

06. Emphasize actionable insights and policy recommendations in the conclusion of the abstract.

Authors’ response: The authors appreciate the reviewer's issues and we have included them as per your recommendation from line numbers 57 to 60.

Introduction:

07. The introduction highlights the problem very well but does not provide a clear articulation of the research gap. For instance, while it mentions previous studies on small samples or in specific regions, how this larger dataset adds value needs to be better stated.

Authors’ response: we thank you for the very valuable comments in this regard and the authors as corrected as suggested by the reviewer from line numbers 92 to 93.

Citations are irregularly formatted and need to follow the journal's formatting rules.

Authors’ response: we thank you for the very valuable comments in this regard and the authors as corrected as suggested by the reviewer from line numbers 64 to 91.

Methods:

08. The multilevel modeling approach has been appropriately described, though model selection and handling of missing data need further clarity.

Authors’ response: Thank you very much dear reviewer for this comment. We revised this section from line numbers 139 to 40.

09. On ethics approval, it says, "Not applicable" since the data used are secondary; however, explaining in detail, how the DHS protects anonymity and adheres to the ethical requirements would strengthen this section.

Authors’ response: Thank you very much for the comment. We accept the comment and the authors included the reviewer's idea from line numbers 490 to 491.

10. The size and scope of the data used in the manuscript require an explanation of how missing data were treated.

Authors’ response: Thank you very much dear reviewer for this comment. We revised this section from line numbers 139 to 40.

11. Although many variables are represented, there is little explanation of why certain predictors were chosen and how they relate to the existing theories or frameworks on solid fuel use. The operationalization of the variables, particularly those at the community level, is not fully described about poverty and education levels.

Authors’ response: Thank you very much for the issues. The proportion of households is categorized as low poverty level (those with ≥ 50%) and higher poverty level (those with <50%) using the national median value. As well as community-level education, the same way as the poverty level is defined. Therefore, we included it in the manuscript from line numbers 134 to 138.

Results:

12. The prevalence of the use of solid fuel is presented quite clearly; tables and figures should be better formatted for readability.

Author’s response: We, the authors would thank you for pointing us to such very relevant comments and these were paraphrased in all tables and figures.

13. Results must integrate findings into contextual interpretations and not merely state statistical outputs. For instance, the discussion of the Adjusted Odds Ratios (e.g., AOR = 12.46 for poor households) could include real-world implications.

Authors’ response: Thank you very much for the comment. Corrected as suggested by the reviewer from line numbers 234 to 285.

14. Table heads and figure captions should be intelligible independently.

Authors’ Response: Thank you for your insightful comment. We have revised the table heads and figure captions to ensure they are self-explanatory considering all tables and figures.

15. The magnitude of the odds ratios seems plausible, but the authors could interpret the outliers or very high AOR values-for example, AOR = 144.49 for community-level poverty critically. Such extreme values are indicative of highly stratified populations or possibly overlapping variables.

Authors’ Response: Thank you for your valuable feedback and authors include the possible reason from line numbers 271 to 278.

16. The AUC is well described but could be briefly explained for readers who may not be familiar with this measure.

Authors’ response: we thank you for this appreciated comment on these issues and we modified as suggested by the reviewer from line numbers 291 to 295.

17. The odds ratio for low-income communities seems very high (AOR = 144.49), questionable about the granularity of the poverty measure. The author should check the causes.

Authors’ Response: Thank you for your valuable feedback and the authors include possible causes of such high AOR from line numbers 271 to 278.

18. The study has shown that even the most complete model- Model III- explains only a small proportion of the variance ICC = 0.37. The authors should present possible unmeasured variables.

Authors’ Response: Thank you for your insightful comment and authors included possible unmeasured variables from line numbers 311 to 314.

Discussion:

19. While the discussion mentions broad recommendations, it lacks specificity. The authors miss an opportunity to provide evidence-based, region-specific recommendations for policymakers.

Authors’ Response: Thank you for your valuable feedback. We believe this will strengthen the overall impact of our work and provide actionable insights for local decision-makers. Therefore, the authors include this pertinent factor from line numbers 447 to 456.

20. While the discussion mentions broad recommendations, it lacks specificity.

Authors’ Response: We appreciate the reviewer’s feedback and we enhanced the discussion by including specific, actionable recommendations tailored to the study findings to provide greater clarity and practical value from line numbers 447 to 456.

21. While the authors recognize the structural and cultural barriers, actionable recommendations as specific programs or policies aimed at reducing dependence on solid fuels are in need of more emphasis in this section.

Authors’ Response: We have incorporated these recommendations in the revised section to ensure actionable and context-specific solutions are adequately emphasized. We trust this enhancement strengthens the manuscript and aligns with your suggestions from line numbers 444 to 456.

22. Authors should critically review the limitations regarding the use of secondary data from DHS, including possible biases and inability to control the selection of variables.

Authors’ Response: Thank you for the insightful comment. We acknowledge that using secondary data from the Demographic and Health Surveys (DHS) has inherent limitations, which we have now addressed more explicitly in the revised manuscript from line numbers 432 to 443.

23. The reported prevalence of solid fuel use is identified and described in the result section of this study which befits the large dataset that this study leveraged. Nevertheless, the presentation might have been contextual, for instance, comparing it with certain countries outside of Sub-Saharan Africa or global averages.

Authors’ Response: Thank you for your thoughtful suggestion. We appreciate your emphasis on contextualizing the reported prevalence of solid fuel use. While the current study primarily focuses on trends and determinants of solid fuel use within Sub-Saharan Africa (SSA), we agree that comparing these findings with countries outside SSA or global averages would provide additional valuable context. We believe this addition will enhance the contextual value of the study and appreciate your valuable input in improving our work from line numbers 324 to 331.

24. While household heads of older age were more likely to use solid fuel, it was specifically with the categories of age >60 years, AOR = 1.12. The study speculates on generational differences in energy preferences; however, the discussion fails to provide empirical evidence or citations to support this claim.

Authors’ Response: Thank you for this interesting comment we included this revision to address the reviewer’s concerns from line numbers 352 to 357.

25. The authors miss an opportunity to provide evidence-based, region-specific recommendations for policymakers.

Authors’ Response: we thank you, dear reviewer, accept the suggestion, and include region-specific, evidence-based recommendations for policymakers in the revised manuscript from line numbers 466 to 473.

26. Through the discussion mentioned the findings of other studies however, it misses the comparison or contrast of findings.

Authors’ response: Authors thank you, dear reviewer; we have compared it with previous studies even if it includes country level from line numbers 326 to 331, but put as a limitation

27. Whereas the authors have focused on individual-level predictors, community-level factors i.e., poverty and education remain underexplored in the discussion. The discussion may be further strengthened by exploring the dynamics of how community-level factors shape or influence individual decisions.

Authors’ response: Authors thank you, dear reviewer; we have included strengthening the manuscript from line numbers 425 to 431.

28. While limitations are recognized, they are only superficially discussed. The study does not discuss the possible bias from using self-reported data or the limitations of secondary data. The greatest limitation is the cross-sectional nature of the analysis, which should be discussed more critically. The longitudinal data might have shown causality, which is missing here.

Authors’ Response: Thank you, dear reviewer, we have incorporated these limitations in the revised manuscript from line numbers 432 to 443.

Conclusion:

29. The conclusion summarizes the findings appropriately but should highlight the study's novelty and its practical implications, such as recommendations for policymakers.

Authors’ Response: we thank you, dear reviewer, accept the suggestion, and include region-specific, evidence-based recommendations for policymakers in the revised manuscript from line numbers 458 to 472.

Overall comment:

The manuscript needs a lot of work in terms of writing quality, contextualization of findings, and depth of discussion.

Authors’ Response: Thank you for your feedback; I will revise the manuscript to improve the writing quality, better contextualize the findings, and deepen the discussion.

Reviewer #2: The following are my comments regarding the paper:

1) The title of the paper seems misleading, as the primary focus appears to be on the determinants of solid fuel usage in sub-Saharan Africa (SSA). The title should better reflect the main objectives and scope of the study.

Authors’ Response: We appreciate the reviewer's comment and have revised the title to more accurately reflect the study's primary focus on the determinants of solid fuel usage in Sub-Saharan Africa from line numbers 1 to 2.

2) The introduction/background and contribution/value addition of that study failed to generate significant interest. The authors should present more compelling points to address these problems.

Authors’ Response: Thank you for your comment we have revised the introduction to emphasize the significance of the study by clearly outlining the research gaps, highlighting its innovative approach, and demonstrating its potential impact in addressing the identified problems from line numbers 96 to 100.

3) The paper is missing a literature review section. The authors should discuss prior studies in the field, identify gaps in the existing literature, and justify how their approach addresses these gaps. This will provide a stronger foundation for the research.

Authors’ Response: We sincerely thank the reviewer for their valuable feedback and insightful comments, which have significantly improved the quality and clarity of our manuscript. So we have revised this part from line numbers 85 to 91.

4) The rationale for using the multilevel logistic regression model is not sufficiently explained. The authors should provide a thorough justification for why this model is appropriate for the study, supported by relevant literature.

Authors’ Response: Thank you for this valuable comment and we have clarified this in the revised manuscript from line numbers 142 to 154.

5) The Demographic and Health Surveys (DHS) were conducted in different years for various countries in SSA. The authors should clearly explain how they combined data from different households and years into a single dataset. They must address potential issues that may arise from merging datasets of varying sizes and discuss why this approach does not compromise the validity of the analysis.

Authors’

---

## [Decision Letter · Decision Letter 2]

27 Feb 2025

PONE-D-24-55055R2Determinants of Solid Fuel Use in Sub-Saharan Africa: A Multilevel Analysis Using DHS DataPLOS ONE

Dear Dr. Azanaw,

Thank you for submitting your manuscript to PLOS ONE. After careful consideration, we feel that it has merit but does not fully meet PLOS ONE’s publication criteria as it currently stands. Therefore, we invite you to submit a revised version of the manuscript that addresses the points raised during the review process. Reviewer 2 has some outstanding corrections on the statistics performed and the policy recommendations.

We look forward to receiving your revised manuscript.

Kind regards,

Alison Parker

Academic Editor

PLOS ONE

Reviewers' comments:

Reviewer's Responses to Questions

**Comments to the Author**

1. If the authors have adequately addressed your comments raised in a previous round of review and you feel that this manuscript is now acceptable for publication, you may indicate that here to bypass the “Comments to the Author” section, enter your conflict of interest statement in the “Confidential to Editor” section, and submit your "Accept" recommendation.

Reviewer #1: All comments have been addressed

2. Is the manuscript technically sound, and do the data support the conclusions?

Reviewer #1: Yes

3. Has the statistical analysis been performed appropriately and rigorously? 

Reviewer #1: Yes

4. Have the authors made all data underlying the findings in their manuscript fully available?

Reviewer #1: Yes

5. Is the manuscript presented in an intelligible fashion and written in standard English?

Reviewer #1: No

6. Review Comments to the Author

Reviewer #1: The authors have made significant revisions based on reviewer feedback, improving the structure, clarity, and scientific rigor of the study. However, few revisions commented below will further enhance the quality of this manuscript.

Comments:

01. The manuscript still contains awkward phasing and grammatical errors, so it requires thorough proofreading by a native speaker.

02. There is still considerable ambiguity surrounding the display of P-values and Adjusted Odds Ratios (AOR). Explain briefly why the odds ratios for some factors are so high or low. Describe how model selection was influenced by AIC, BIC, and ICC values.

03. The policy suggestions are still general and wide-ranging. Provide evidence-based, nation-specific solutions to reinforce the policy implications.

7. PLOS authors have the option to publish the peer review history of their article (what does this mean? ). If published, this will include your full peer review and any attached files.

**Do you want your identity to be public for this peer review?** For information about this choice, including consent withdrawal, please see our Privacy Policy .

Reviewer #1: No

---

## [Author Response · Author response to Decision Letter 3]

28 Feb 2025

A point-by-point response to editor and reviewers

Please accept our revised manuscript and note our point-by-point response to the reviewer below for the manuscript titled “PONE-D-24-55055R2

Determinants of Solid Fuel Use in Sub-Saharan Africa: A Multilevel Analysis Using DHS Data

PLOS ONE”

Note: Responses for each issue raised by the Academic Editor and Reviewers are indicated by highlight.

Authors’ response: Thank you, dear editor. We have seen all your concerns and have done as per your recommendation.

Reviewer #1: The authors have made significant revisions based on reviewer feedback, improving the structure, clarity, and scientific rigor of the study. However, the few revisions commented on below will further enhance the quality of this manuscript.

Comments:

01. The manuscript still contains awkward phrasing and grammatical errors, so it requires thorough proofreading by a native speaker.

Authors’ response: Thank you very much, dear reviewer, for your comments that were able to fill the authors' gaps. Therefore, the authors rephrased and edited the whole document.

02. There is still considerable ambiguity surrounding the display of P-values and Adjusted Odds Ratios (AOR). Explain briefly why the odds ratios for some factors are so high or low. Describe how model selection was influenced by AIC, BIC, and ICC values.

Authors’ response: Thank you very much, dear reviewer. We have explanations of P-values along with the AOR from line numbers 154 to 169.

03. The policy suggestions are still general and wide-ranging. Provide evidence-based, nation-specific solutions to reinforce the policy implications.

Authors’ response: Authors thank you, dear reviewer; we have revised from line number 366.

Authors’ response: We believe that we have addressed all the issues raised by the dear editor and reviewers sufficiently. If there are additional issues the reviewer expects to be addressed, we encourage further elaboration on the issues. Thank you once again for the valuable comments raised, the knowledge sharing, and your time to help us.

---

## [Editor Report · Decision Letter 3]

12 Mar 2025

Determinants of Solid Fuel Use in Sub-Saharan Africa: A Multilevel Analysis Using DHS Data

PONE-D-24-55055R3

Dear Dr. Azanaw,

We’re pleased to inform you that your manuscript has been judged scientifically suitable for publication and will be formally accepted for publication once it meets all outstanding technical requirements.

Kind regards,

Alison Parker

Academic Editor

PLOS ONE
---

## [Editor Report · Acceptance letter]

PONE-D-24-55055R3

PLOS ONE

Dear Dr. Azanaw,

I'm pleased to inform you that your manuscript has been deemed suitable for publication in PLOS ONE. Congratulations! Your manuscript is now being handed over to our production team.

Kind regards,

on behalf of

Dr. Alison Parker

Academic Editor

PLOS ONE